# EFFICIENT SCALING OF DIFFUSION TRANSFORMERS FOR TEXT-TO-IMAGE GENERATION

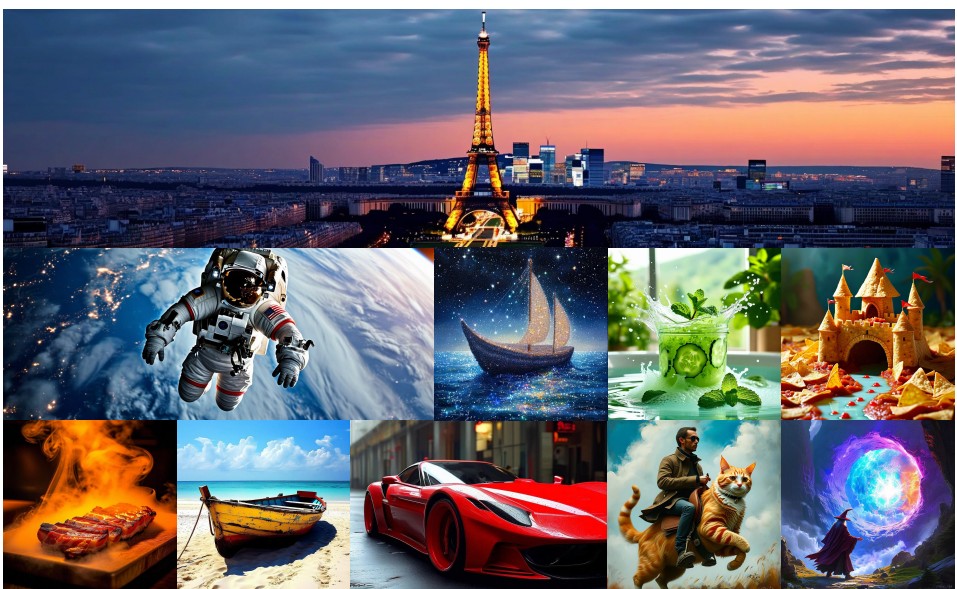

Figure 1: Examples of high-resolution images generated by a 2.3B U-ViT 1K model.

## ABSTRACT

We empirically study the scaling properties of various Diffusion Transformers (DiTs) for text-to-image generation by performing extensive and rigorous ablations, including training scaled DiTs ranging from 0.3B upto 8B parameters on datasets up to 600M images. We find that U-ViT, a pure self-attention based DiT model provides a simpler design and scales more effectively in comparison with cross-attention based DiT variants, which allows straightforward expansion for extra conditions and other modalities. We identify a 2.3B U-ViT model can get better performance than SDXL UNet and other DiT variants in controlled setting. On the data scaling side, we investigate how increasing dataset size and enhanced long caption improve the text-image alignment performance and the learning efficiency.

## 1  INTRODUCTION

Transformer (Vaswani et al., 2017)'s straightforward design and ability to scale efficiently has driven significant advancements in large language models (LLMs) (Kaplan et al., 2020). Its inherent simplicity and ease of parallelization makes it well-suited for hardware acceleration. Diffusion Transformers (DiTs) (Peebles & Xie, 2023; Bao et al., 2023) initially replaces UNet with transformers for diffusion-based image generation results in the proposal of numerous variants (Chen et al., 2024b; Esser et al., 2024b; Crowson et al., 2024; Gao et al., 2024; Li et al., 2024b) and has since successfully expanded into video generation (Brooks et al., 2024).

Despite the rapid evolution of DiT models, a comprehensive comparison between various DiT architectures and UNet-based models for text-to-image generation (T2I) is still lacking. The impact of model design on DiT's ability to accurately translate text descriptions into images (text-to-image

alignment) remains unclear. Furthermore, the optimal scaling strategy for transformer models in T2I tasks compared to UNet is yet to be determined. The challenge of establishing a fair comparison is further compounded by the variation in training settings and the significant computational resources required to train these models.

In this work, we empirically study the scaling properties of several representative DiT architectures for T2I by performing rigors ablations, including training scaled DiTs ranging from 0.3B to 8B parameters on datasets up to 600M images in controlled settings. Specifically, we ablate and scale three DiT variants, i.e., PixArt-$\alpha$ (Chen et al., 2024b), LargeDiT (Gao et al., 2024), and U-ViT (Bao et al., 2023). We train them from scratch on large-scale datasets without using pre-trained DiT initialization or ImageNet pre-training. All DiT variants are trained in a controlled setting with the same autoencoder, text encoder and training settings for fair comparison. We find that U-ViT's simpler architecture design facilitates efficient model scaling and supports image editing by simply expanding condition tokens. Finally, we explore the impact of dataset scaling, considering both dataset size and caption density. The main contributions of our work include:

- We compare the architecture design of three text conditioned DiT models including scaled PixArt-$\alpha$, LargeDiT and U-ViT variants in controlled settings, allowing a fair comparison of recent DiT variants for real-world text-to-image generation.

- We scale the three DiT variants along depth and width dimensions and verify their scalability with model size as large as 8B. We find that the U-ViT architecture, a full self-attention based ViT with skip connections, has competitive performance with other DiT designs. The scaled 2.3B U-ViT can outperform SDXL's UNet and much larger PixArt-$\alpha$ and LargeDiTs.

- We verified that the full self-attention design of U-ViT allows training image editing model by simply concatenating masks or condition image as condition tokens, which shows better performance than traditional channel concatenation approach.

- We examined why long caption enhancement and dataset scaling help to improve the text-image alignment performance. We find captions with higher information density can yield better text-image alignment performance.

## 2 RELATED WORK

**Transformers for T2I** U-Net (Ronneberger et al., 2015) was the de facto standard backbone for diffusion based image generation since (Ho et al., 2020) and is widely used in text-to-image models including LDM (Rombach et al., 2021), SDXL (Podell et al., 2023), DALL-E (Ramesh et al., 2022) and Imagen (Saharia et al., 2022). U-ViT (Bao et al., 2023) treats all inputs including the time, condition and noisy image patches as tokens and employs ViT equipped with long skip connections between shallow and deep layers, suggesting the long skip connection is crucial while the downsampling and up-sampling operators in CNN-based U-Net are not always necessary. DiT (Peebles & Xie, 2023) replaces U-Net with Transformers for class-conditioned image generation and identify there is a strong correlation between the network complexity and sample quality. PixArt-$\alpha$ (Chen et al., 2024b) extends DiT (Peebles & Xie, 2023) for text-conditioned image generation by initializing from DiT pre-trained weights. PixArt-$\Sigma$ (Chen et al., 2024a) upgrades PixArt-$\alpha$ with stronger VAE, larger dataset and longer text token limit. It introduces token compression to support 4K image generation. Those DiT variants are mostly around 0.6B and focus on showing comparable results with U-Net.

**Scaling DiTs** HourglassDiT (Crowson et al., 2024) introduces hierarchical design with down/up sampling in DiT and reduces the computation complexity for high resolution image generation. SD3 (Esser et al., 2024b) presents a transformer-based architecture that uses separate weights for the image and text modalities and enables a bidirectional flow of information between image and text tokens. They parameterize the size of the model in terms of the model's depth and scale up the backbone to 8B. Large-DiT (Gao et al., 2024) incorporates LLaMA's text embedding (Touvron et al., 2023) and scales the DiT backbone. Specifically, they modify the causal attention of LLaMA to a bidirectional attention mechanism. They normalize the key and query within the attention mechanism. They show scaling-up parameters up to 7B can improve the convergence speed. They further extend it for generating multiple modalities in flow-based Lumnia-T2X (Gao et al., 2024) and Lumina-Next (Zhuo et al., 2024).

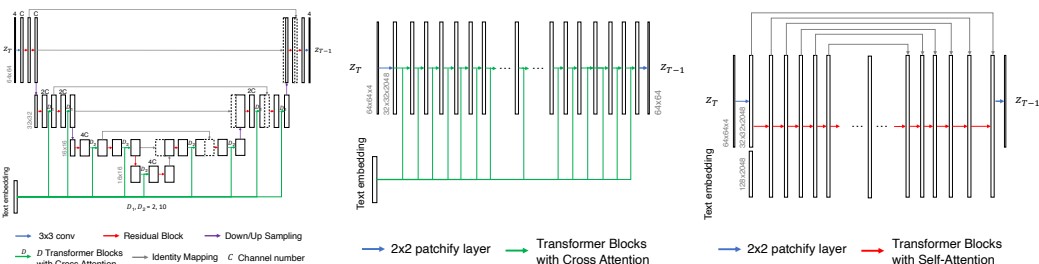

Figure 2: Illustration of the design of SDXL U-Net, DiT (e.g., PixArt-$\alpha$/LargeDiT) and U-ViT.

# 3 SCALING DIFFUSION TRANSFORMERS

We ablate and scale three DiT variants, i.e., PixArt-$\alpha$ (Chen et al., 2024b), LargeDiT (Gao et al., 2024), and U-ViT (Bao et al., 2023) in controlled settings. Fig. 2 compares the archtiecture design among U-Net, DiT and U-ViT. To fairly compare DiT variants with U-Net models, we replace SDXL's U-Net with the DiT backbones and keep other components the same, i.e., we use SDXL's VAE and OpenCLIP-H (Ilharco et al., 2021) text encoder.

## 3.1 ABLATION SETTINGS

**Model Design Space**    We ablate the transformer-based diffusion model design in the following dimensions: 1) hidden dimension $h$: we scale it from 1024 to 3072. 2) transformer depth $d$: we scale the transformer blocks from 28 to 80. 3) number of heads $n$, we keep it fixed to 16 or 32.

**Training Settings**    We mainly train models and perform ablations on our curated dataset *LensArt*, which contains 250M text-image pairs. For additional data scaling experiment in later sections, we also use our curated dataset *SSTK*, which contains 350M text-image pairs. We train all models at 256×256 resolution with batch size 2048 up to 600K steps. We follow the setup of LDM (Rombach et al., 2021) for DDPM schedules. We use AdamW (Loshchilov & Hutter, 2019) optimizer with 10K steps warmup and then constant learning rate $8e^{-5}$. We employ mixed precision training with BF16 precision and enable FSDP (Zhao et al., 2023) for large models.

**Evaluation**    We use the DDIM sampler (Song et al., 2020) in 50 steps with fixed seed and CFG scale (7.5) for inference. We follow the setting of (Li et al., 2024a) for evaluation on composition ability and image quality with: 1) **TIFA** (Hu et al., 2023) measures the faithfulness of a generated image to its text input via VQA. It contains 4K collected prompts and corresponding question-answer pairs generated by a language model. Image faithfulness is calculated by checking whether existing VQA models can answer these questions using the generated image. 2) **ImageReward** (Xu et al., 2023) was learned to approximates human preference. We calculate the average ImageReward score over images generated with sampled 10K MSCOCO (Lin et al., 2014) prompts. More evaluation metrics can be found in Appendix D.

## 3.2 SCALING PIXART-$\alpha$

Previous study (Li et al., 2024a) scales PixArt-$\alpha$ (Chen et al., 2024b) from 0.5B to 1.1B to compare with similar sized U-Nets. They find that similar sized PixArt-$\alpha$ performs worse than U-Net. We followed their setting and further scaled PixArt-$\alpha$ upto 3.0B from both depth and width dimensions. For depth scaling, we fix $h$ at 1024 and 1536 while changing $d$ from 28 to 80. For width scaling, we fix $d$ to 28 and 42 while changing $h$ from 1152 to 2048. Fig. 3 shows how TIFA score scales along depth and width dimensions. All PixArt-$\alpha$ variants yield lower TIFA and ImageReward scores in comparison with SD2 U-Net trained in same steps, e.g., SD2 U-Net reaches 0.80 TIFA at 250K steps while the 0.9B PixArt-$\alpha$ variant gets 0.78. Chen et al. (2024b) also report that training without ImageNet pre-training tends to generate distorted images in comparison to models initialized from pre-trained DiT weights, which is trained 7M steps on ImageNet. Though Chen et al. (2024b) proves that U-Net is not a must for diffusion models, PixArt-$\alpha$ variants do take longer iterations and

more compute to achieve similar performance as U-Net. The 3B PixArt-$\alpha$ model still cannot match SD2-U-Net within same training steps.

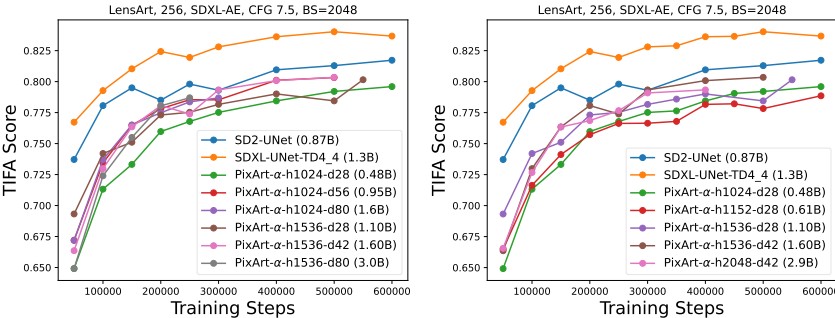

Figure 3: Scaling PixArt-$\alpha$ on the depth and width dimensions.

### 3.3 SCALING LARGEDIT

We further employ LargeDiT (Gao et al., 2024) as the denoising backbone and explore its scaled version. The original LargeDiT comes with 0.6B, 3B[1], and 7B pre-trained versions. We ablate LargeDiT in the dimension of depth, hidden dimension, and number of heads. As shown in Fig. 4, the 1.7B LargeDiT-$h1536$-$d42$-$n32$ is on par with SD2 U-Net with 0.80 TIFA. The LargeDiT models start to surpass SD2 U-Net since the 2.9B model ($h2048$-$d42$-$n32$). The 4.4B variants shows close metrics to SDXL TD4-4 U-Net at 600K steps. However, further enlarging the model does not improve the performance. The 7.6B model variant gets similar performance as the 4.4B version, and there is still a gap with SDXL U-Net.

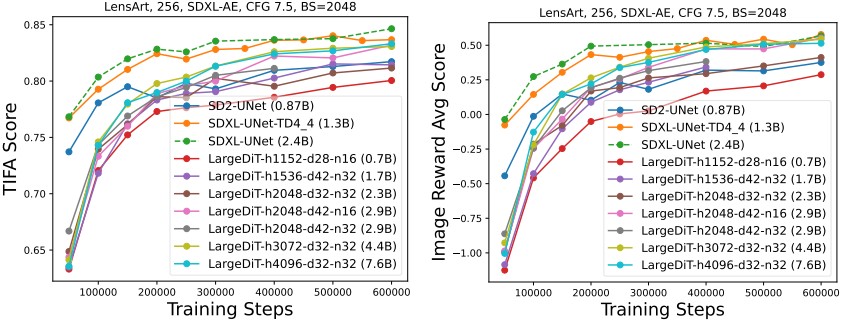

Figure 4: Scaling LargeDiT variants from 1.7B to 7.6B.

### 3.4 SCALING U-VITS

**Scaling the hidden dimension and layer depth**   We train scaled U-ViT variants to see whether we can get better performance than U-Net. We start from the original 0.6B version ($h1152$-$d28$-$n16$) and scale along both depth and width dimensions to get the 1.3B version ($h1536$-$d42$-$n16$). As shown in Fig. 5, further increasing hidden dimension $h$ to 2048 results in the 2.3B U-ViT model ($h2048$-$d42$-$n16$), which shows significantly better performance than SD2 U-Net and matches SDXL U-Net in both TIFA and ImageReward after 500K steps. To further scale the backbone, we increased $h$ to 2560 and $d$ to 56 on top of the 2.3B model, resulted in the 3.1B and 3.7B model. However, we did not observe significantly improved performance. Table. 1 shows detailed configurations. We can see that the 2.3B U-ViT's inference latency is 75% and 66% less than SDXL U-Net at 256 and 512 resolution respectively, though its thoretical GMACs is 3× more.

**Comparison with PixArt-$\alpha$ and LargeDiT at Different Scales**   The major difference among different DiT variants lie in the block design and the integration of text conditioning. Here we

---

[1]We find it occupies 4.4B in our setting, and aligns with LuminaT2X's 5B setting.

Table 1: Sampling space for scaling U-ViT and their inference cost at different resolutions. The latency is end-to-end inference time (s) with DDIM 50 steps on H100 GPUs, the relative latency (%) is compared with SDXL. All models use the same VAE, text encoder and patch size 2.

| Model | $h$ | $d$ | $n$ | Params (B) | 256x256 | | | 512x512 | | | 1024x1024 | | |
|---|---|---|---|---|---|---|---|---|---|---|---|---|---|
| | | | | | TMACs | Latency | % | TMACs | Latency | % | TMACs | Latency | % |
| SD2 (Rombach et al., 2021) | | | | 0.9 | 0.09 | 1.81 | 0.34 | 1.35 | 3.31 | 0.60 | 1.35 | 3.22 | 0.53 |
| SDXL-TD4-4 (Li et al., 2024a) | | | | 1.3 | 0.14 | 3.28 | 0.62 | 0.55 | 3.7 | 0.67 | 2.18 | 4.12 | 0.67 |
| SDXL (Podell et al., 2023) | | | | 2.4 | 0.20 | 5.30 | 1.00 | 0.75 | 5.52 | 1.00 | 2.98 | 6.12 | 1.00 |
| PixArt-$\alpha$ (Chen et al., 2024b) | 1152 | 28 | 16 | 0.6 | 0.14 | 1.76 | 0.33 | 0.54 | 1.77 | 0.32 | 2.14 | 4.32 | 0.71 |
| LargeDiT-5B (Gao et al., 2024) | 3072 | 32 | 32 | 4.4 | 0.11 | 2.24 | 0.42 | 3.84 | 2.85 | 0.52 | 15.09 | 10.97 | 1.79 |
| LargeDiT-7B (Gao et al., 2024) | 4096 | 32 | 32 | 7.6 | 1.90 | 2.23 | 0.42 | 6.86 | 4.17 | 0.76 | 26.96 | 16.22 | 2.65 |
| U-ViT-Large (Bao et al., 2023) | 1024 | 20 | 16 | 0.3 | 0.10 | 0.68 | 0.13 | 0.31 | 0.76 | 0.14 | 1.19 | 2.26 | 0.37 |
| U-ViT-Huge (Bao et al., 2023) | 1152 | 28 | 16 | 0.5 | 0.17 | 0.99 | 0.19 | 0.55 | 1.02 | 0.18 | 2.08 | 4.17 | 0.68 |
| Scaled U-ViTs | 1536 | 42 | 16 | 1.30 | 0.44 | 1.25 | 0.24 | 1.45 | 1.35 | 0.24 | 5.50 | 6.76 | 1.10 |
| | 2048 | 32 | 16 | 1.8 | 0.60 | 0.98 | 0.18 | 1.98 | 1.44 | 0.26 | 7.49 | 6.78 | 1.11 |
| | **2048** | **42** | **16** | **2.3** | 0.78 | 1.31 | 0.25 | 2.58 | 1.85 | 0.34 | 9.77 | 8.60 | 1.41 |
| | 2048 | 64 | 16 | 3.6 | 1.18 | 1.89 | 0.36 | 3.90 | 2.79 | 0.51 | 14.78 | 13.20 | 2.16 |
| | 3072 | 32 | 32 | 4.0 | 1.35 | 1.11 | 0.21 | 4.45 | 2.72 | 0.49 | 16.86 | 15.40 | 2.52 |
| | 3072 | 42 | 32 | 5.3 | 1.76 | 1.3 | 0.25 | 5.80 | 3.53 | 0.64 | 21.98 | 20.32 | 3.32 |
| | 3072 | 48 | 32 | 6.0 | 2.00 | 1.49 | 0.28 | 6.61 | 4.01 | 0.73 | 25.00 | 22.80 | 3.73 |
| | 3072 | 64 | 32 | 8.0 | 2.66 | 1.98 | 0.37 | 8.78 | 5.30 | 0.96 | 33.25 | 30.00 | 4.90 |

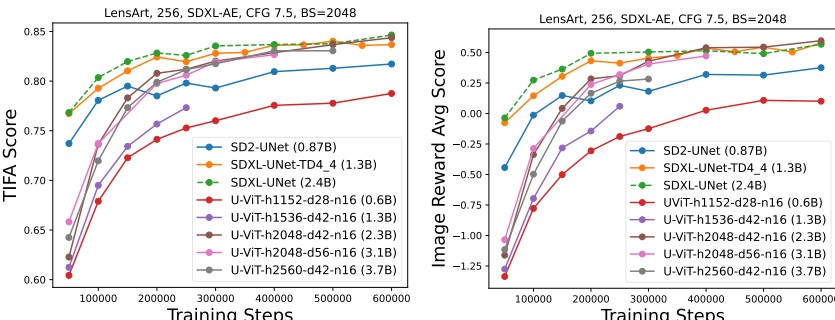

Figure 5: Scaling U-ViT along hidden dimension $h$, depth $d$ and combined dimensions on LensArt.

compare PixArt-$\alpha$, LargeDiT and U-ViT in similar architecture settings. Specifically, we compare them in the configurations of original DiT-XL (0.6B) and their scaled versions at the 2B level. Fig. 6 (a) shows that at small architecture scales (0.6B), the U-ViT model converges slower but still results in competitive or better results at later stage. When the model scales both hidden dimension and depth to parameter size at 2B level, the U-ViT model converges faster than LargeDiT and PixArt-$\alpha$ models. We conjecture the difference lies in how the textual information is processed by the diffusion models. For DiT models the textual condition is passed at all layers and is processed by a cross-attention layer. However, for U-ViT, the textual information is only passed once in the first layer along with the image patches, and is then processed by the transformer. We observe in Fig. 6 that for larger latent dimension, the refinement of textual information by the U-ViT is essential for scaling as it enables the model to outperform PixArt-$\alpha$ and LargeDiT. We will verify this conjecture in Sec 4.2.

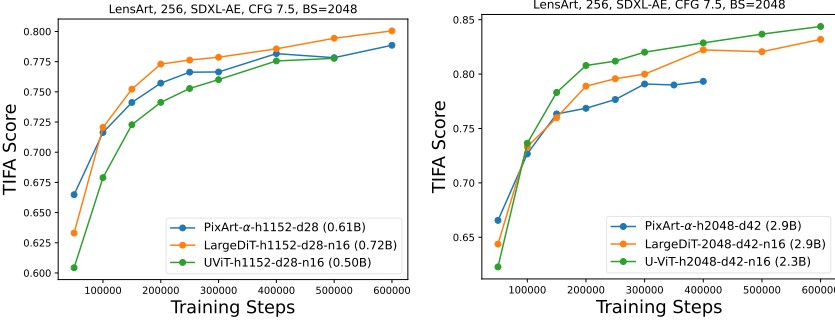

Figure 6: Comparing different DiT designs in similar architecture hyperparameters at different scales.

## 4 ABLATING U-ViTs

To understand why scaled U-ViT outperforms U-Net, PixArt-$\alpha$ and LargeDiT variants, we first analyze the design of U-Net and U-ViT, and then ablate the effect of text encoder fine-tuning.

### 4.1 COMPARING U-NET AND U-ViT

The major difference between U-Net and U-ViT lies at that: 1) U-Net has down/up sampling layers, the text embedding is sent to every spatial Transformer blocks; there are skip connections connecting input and output blocks. 2) U-ViT has no down/up sampling layers. The condition tokens are contacted with timestep embedding at the beginning of the input. Bao et al. (2023) show that the long skip connection is crucial, while the downsampling and up-sampling operation in CNN-based U-Net are not always necessary.

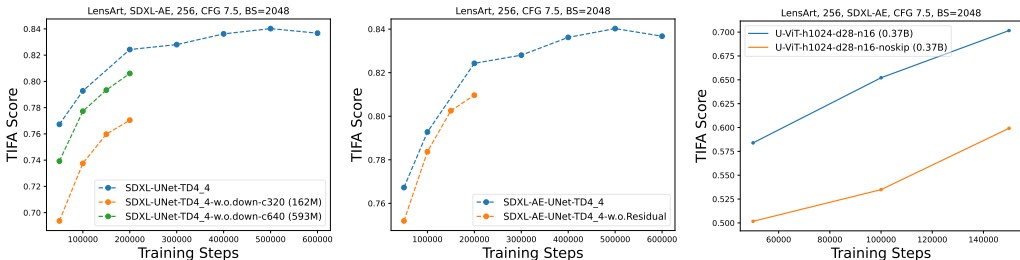

Figure 7: (a) The effect of removing downsampling and using fixed channels in U-Net. (b) The effect of skip connection in U-Net. (c) The effect of skip connection in U-ViT.

**The Effect of Down/Up sampling in U-Net**    To make U-Net more like transformers, we remove the down/up sampling layers in U-Net and fix the numer of channels per layer. As shown in Fig. 7(a), the consistent channel size (320) without down/up sampling results in a much smaller model (162M) and worse performance. Further increasing the initial channels (from 320 to 640) significantly improves the performance, which indicates the importance of channel number (width) for U-Net.

**Skip Connections in U-Net and U-ViT**    We ablate the effect of skip connections in a smaller version of SDXL with 1.3B parameters (SDXL-TD4_4 (Li et al., 2024a)). As shown in Fig. 7(b), removing residual connection has slower convergence than original U-Net, which implies the importance of skip connection in U-Net. To ablate the effect of skip connections in U-ViT, we first train a small U-ViT with depth 28 as PixArt-$\alpha$ with hidden dimension 1024. We also train the same U-ViT but with the skip connection disabled. Fig. 7(c) verifies that the importance of skip connections in U-ViT. Note that the in-context conditioning scheme in DiT (Peebles & Xie (2023)) is similar to the self-attention in U-ViT. While Peebles & Xie (2023) show inferior performance of in-context conditioning comparising with the cross-attention design, the success of U-ViT indicates the importance of skip connection to make in-context conditioning working.

### 4.2 SELF-ATTENTION AS FINE-TUNING TEXT ENCODERS

In this section, we explore how fine-tuning text encoder impacts the performance for cross-attention and self-attention based models. Currently all T2I models  (Podell et al., 2023; Ramesh et al., 2022; Saharia et al., 2022; Bao et al., 2023; Chen et al., 2024b) keep the text encoder frozen during training. And cross-attention based backbones like UNet, PixArt-$\alpha$ and LargeDiT cross-attend the text embeddings with the latent visual tokens, where the text tokens are fixed in each transformer block. U-ViT concatenates the text tokens with image tokens and passes them through a sequence of transformer blocks, where the text conditioning tokens are modified after each transformer block. The transformer blocks in U-ViT can therefore be *implicitly considered as part of text encoder which is being fine-tuned*.

We empirically test the hypothesis by training cross-attention based models and self-attention based models on LensArt with frozen and non-frozen text encoder. We also ablate the effect of different text encoders when fine-tuning is enabled. We train the denoising backbone using a learning rate

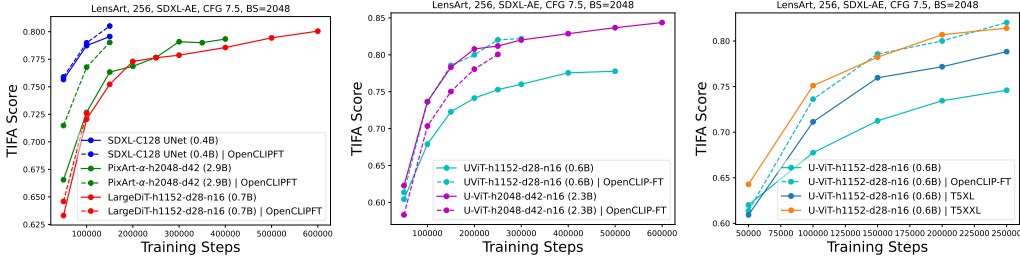

Figure 8: (a) Comprising fine-tuning and freezing text encoder during training for cross-attention based models including UNet, PixArt-$\alpha$ and LargeDiT. (b) Fine-tuning and freezing text encoder with 0.6B and 2.3B U-ViT. (c) Training the 0.6B U-ViT with fixed/trainable OpenCLIP-H and stronger text-encoders including T5XL and T5XXL. Fine-tuning OpenCLIP-H achieves similar performance as using fixed T5XXL.

of 8e-5, and fine-tune the text encoder with a lower learning rate of 8e-6 and weight decay of 1e-4 to prevent from overfitting and diverging too much from the original weights. Fig. 8(a) shows that fine-tuning text encoder results in better performance for all cross-attention based models compared to their frozen variants. For self-attention based U-ViT, we see in Fig. 8(b) that smaller U-ViT can still benefit from fine-tuning the text encoder, while larger U-ViT with frozen text encoder performs the same as fine-tuning the text encoder with a smaller U-ViT. Further fine-tuning the text encoder with large U-ViT does not improve much for large U-ViT models. Fig. 8(c) shows that fine-tuning a weak text encoder (OpenCLIP-H) can achieve similar performance with using a frozen stronger text encoder (T5XXL).

## 5 SCALING THE NUMBER OF TOKENS

### 5.1 SCALING THE NUMBER OF TOKENS FOR HIGH-RESOLUTION TRAINING

With patch size 2, the number of latent image tokens is 1024 for generating $512^2$ images, and it increases to 4096 for $1024^2$ images. As shown in Table 1, the 2.3B U-ViT 1K model having $3.2\times$ more theoretical computation cost than SDXL U-Net but only yields 41% higher end-to-end latency. We find it is critical to adjust the noise scheduling for the 1K resolution training of U-ViT. Using the same noise scheduling as 256 and 512 resolution training leads to background and concept forgetting as well as color issues. This aligns with previous findings on training high-resolution diffusion models (Chen, 2023; Hoogeboom et al., 2023; Esser et al., 2024b). We show examples of high-resolution images in different aspect ratios generated by the 2.3B U-ViT 1K model in Fig 1.

### 5.2 SCALING CONDITION TOKENS FOR IMAGE EDITING

In image generation, conditioning can be applied using both global (e.g., text embeddings, CLIP image embeddings) and local factors (e.g., edge maps, masks). Recent works (Zhao et al., 2024; Ye et al., 2023) have trained models that handle multiple conditions simultaneously; however, these models differentiate between global and local conditions in their handling. Global conditions are typically cross-attended by the noise latents, while local conditions are concatenated with them. This distinction is often a result of the limitations imposed by widely used diffusion backbones like UNet and DiTs. In this section, we show that we can adapt U-ViT to any new condition by just scaling up the tokens - we tokenize the condition and concatenate it with noise and text tokens, without having to incorporate any specialized logic to handle different types of conditions. In contrast to methods that concatenate the condition with noise latents along the channel dimension, our approach does not require the local condition to have the same resolution as the noise latents. We use 2.3B U-ViT pretrained on 1K resolution data for all experiments in this section.

**Image Inpainting via Scaling Condition Tokens:** In text-conditioned image generation, U-ViT concatenates noise latents and text embeddings along the token dimension, enabling self-attention. For adapting U-ViT to the inpainting task, there are two approaches, as shown in Fig 9. The model is trained in the standard manner, optimizing the L2 loss. We use the same datasets as those for text-to-image training, generating masks randomly. We train inpainting models using both methods and evaluate them on ImageReward (for image fidelity) and a modified version of TIFA metric (for

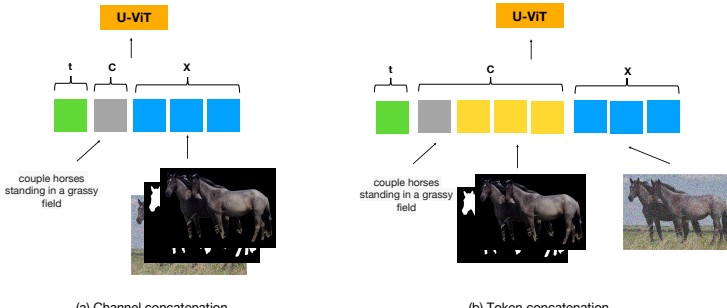

Figure 9: Channel and token concatenation for extending U-ViT to image inpainting. (a) Concatenating the condition image with the noise image along the channel dimension and then tokenizing. This approach maintains the same number of tokens as in T2I generation but requires special handling of the new condition. (b) Tokenizing the new condition (input image + mask), adding positional embedding to it, and concatenating text embeddings along the token dimension.

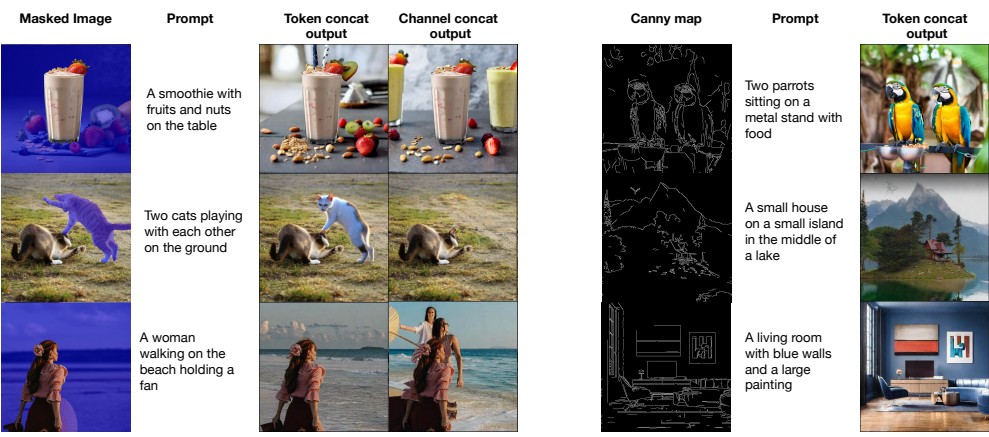

Figure 10: a) Comparison of token and channel concatenation approaches for image inpainting on top of 2.3B 1K resolution pretrained U-ViT: The token concatenation approach demonstrates superior prompt adherence and produces outputs that blend more seamlessly with the surrounding image compared to the channel concatenation method. In row 1, channel concatenation generates more smoothies than the prompt requests. In row 2, channel concatenation does not generate the second cat. In row 3, channel concatenation generates a cluttered output. b) Canny conditioning using token concatenation on top of 2.3B 1K resolution pretrained U-ViT: Qualitative samples showing that the token concatenation approach can handle different types of conditions like canny map.

image-text alignment) which we call TIFA-COCO [2]. The token concatenation method achieves a TIFA-COCO score of 0.887 and an ImageReward of 1.30, outperforming the channel concatenation approach, which achieves a TIFA-COCO score of 0.881 and an ImageReward of 1.24. Token concat scheme allows more fine-grained relationship to be established between noise latent and image condition across transformer blocks through self-attention, ensuring that the network does not lose access to the condition like in channel concat approach. We quantitatively compare our token concatenation approach with current state-of-the-art inpainting method BrushNet (Ju et al., 2024) in Appendix C and find that our approach outperforms BrushNet on TIFA-COCO metric as well as on multiple metrics in BrushBench benchmark (Ju et al., 2024). We show some qualitative outputs on BrushBench dataset in Fig 10 (a).

**Canny Conditioning via Scaling Condition Tokens:** Although Fig 9 illustrates the token and channel concatenation schemes for inpainting, this approach is generalizable to various conditions, such as canny edge maps or segmentation maps. For instance, given a canny edge map, we can adapt U-ViT to condition on it by tokenizing the map, concatenating it with noise and text tokens, and training the model using the standard L2 loss. Unlike methods like ControlNet (Zhang et al., 2023), which require specialized adaptors for processing different conditions, our approach allows for

---

[2]The TIFA and ImageReward scores here are different from the base model. Details in the Appendix C.

conditioning on diverse inputs in a uniform manner. To fine-tune U-ViT efficiently, parameter-efficient fine-tuning (PEFT) methods like LoRA (Hu et al., 2021) can be employed. However, as PEFT is not the focus of our work, we do not explore it further here. We show some qualitative results on canny conditioning using token concatenation in Fig 10 (b). We observe that our proposed approach outperforms Canny ControlNet trained on SD3-Medium (Esser et al., 2024a) on TIFA-COCO and all metrics in BrushBench. We provide this comparison in Appendix C.

## 6 DATA SCALING

### 6.1 CAPTION SCALING: ORIGINAL CAPTION VS. LONG CAPTION

**Generating Long Synthetic Captions** LensArt is curated from web images with alt-text as captions. Thus, the original captions are short, noisy, and often not well-aligned with the image. Inspired by recent works (Chen et al., 2024b; Betker et al., 2023), we use multi-modal LLM to generate long synthetic captions. Concretely, following (Chen et al., 2024b), we use *Describe this image and its style in a very detailed manner* as the prompt to generate long synthetic image captions. Different from (Chen et al., 2024b), we use LLaVA-v1.6-Mistral-7B (Liu et al., 2024b;a; Jiang et al., 2023) to generate long captions on *LensArt* due to its better long captioning performance and lower hallucination compared to other LLaVA variants as benchmarked in THRONE (Kaul et al., 2024), a recent long image caption and hallucination benchmark. We plot the histogram of caption length of original caption and long synthetic caption in Fig. 11, which shows that long synthetic captions contains much more number of tokens compared to original captions.

**Training with Long Captions Scales Better** To study the scaling from short to long captions, we train 0.6B U-ViT on three different datasets: (1) LensArt, which uses original captions and (2) LensArt-Long, which samples original and long captions with equal probability. (3) SSTK with original captions. The results in Fig. 12 show that *LensArt* augmented with long captions provides better results than using original captions as well as SSTK dataset, demonstrating that T2I models scale better with long captions.

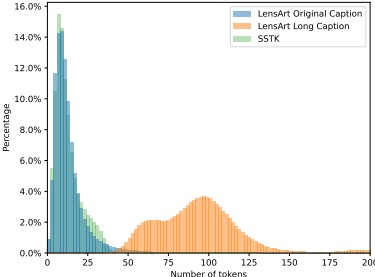

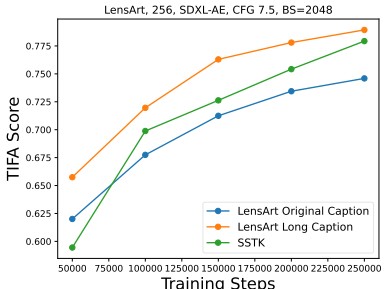

Figure 11: The distribution of caption length (number of tokens) of LensArt original caption, LensArt long caption, and SSTK caption.

Figure 12: TIFA convergence for models trained on LensArt original captions, LensArt long captions, and SSTK captions.

### 6.2 DATA SIZE SCALING

To study the effect of scaling up training data size, we add *SSTK* upon *LensArt* as the training data. As a result, the training data size is scaled from 250M (*LensArt* only) to 600M (*LensArt+SSTK*). Note that we do not use long captions for studying data size scaling. In Fig. 13, we show that the performance in TIFA and Image Reward can be improved as long as the data size is scaled up regardless of the choice of architecture. While previously we show that long captions improve the T2I performance in Sec. 6.1, we find that the distribution of caption length between LensArt and SSTK are very close as shown in Fig. 11. Thus, we conclude that the performance improvement mainly originates from data size scaling.

**U-ViT scales better than UNet with larger data size** Here we compare the data scalability of U-ViT against SDXL. We scale up data size from LensArt (250M) to LensArt + SSTK (600M). The

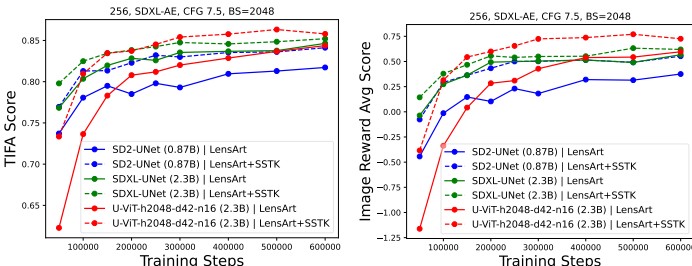

Figure 13: The scalability of U-ViT and SD models trained with larger datasets. Solid lines are on LensArt and dashed lines are on the combination of LensArt and SSTK dataset.

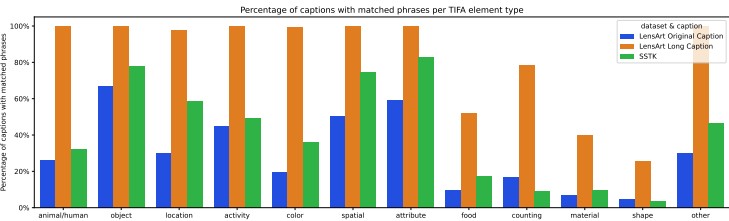

Figure 14: Percentage of captions with matched TIFA element phrases in each TIFA element type. Both LensArt long caption and SSTK caption have higher percentage than LensArt original caption, indicating both of them improve information density compared to LensArt original caption.

results are shown in Fig. 13. While U-ViT performs similar to SDXL on LensArt, U-ViT achieves much larger relative performance gain from *LensArt* to *LensArt+SSTK* compared to SDXL. From another perspective, larger scale of data helps U-ViT scale faster compared to SDXL—while U-ViT does not match performance of SDXL until 400K or 500K training steps (*c.f.* red solid line vs. green solid line in Fig. 13), U-ViT start to surpasses SDXL in 150K steps when data size is scaled up. Overall, the results demonstrate that U-ViT scales better than UNet with larger training data size.

### 6.3 LONG CAPTION AND DATA SCALING INCREASES INFORMATION DENSITY

Since both scaling up long caption and data size can improve text-image alignment, we aspire to unertand the common factor contributing to the improvement. We hypothesize that both LensArt-long and SSTK captions provide more information that can be measured by TIFA. TIFA leverages VQA to answer questions on generated images and measures answer accuracy. Specifically, given an image caption, several pairs of question and answers are generated. Each question corresponds to one element in the caption. Each element belongs to a type (e.g., *surfer* belongs to the *animal/human* element type). TIFA comprehensively evaluates the following element types: *animal/human, object, location, activity, color, spatial, attribute, food, counting, material, shape, other*. Based on element phrases (note that each element can contain more than one English word) in TIFA, we do phrase matching for (1) LensArt original caption, (2) LensArt long caption, and (3) SSTK captions. In Fig. 14, we show the percentage of captions with matched element phrases in each element type. The results show that both LensArt long captions and SSTK have higher percentage of captions with matched TIFA element phrases compared to LensArt caption in almost all TIFA element types, which explains the relative performance difference in Fig. 12. Thus, we conclude that captions with *higher information density* (i.e., not necessarily long caption length) is key to improve text-image alignment.

## 7 CONCLUSION

We performed large-scale ablation of DiT variants and showed the potential scaling ability of different backbone designs. We observe that the architecture like U-ViT that self attends on both condition tokens and image tokens scales more effectively than the cross-attention based DiT variants as measured by TIFA and ImageReward. The design of U-ViT that self attains on all tokens allows straightforward extension for image editing by just expanding condition tokens. It also allows for naive extension to other modality generation, such as text-to-video generation.

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

## A    DATASET DETAILS

We use two proprietary datasets named LensArt and SSTK (Li et al., 2024a). *LensArt* consists of 250 million image-text pairs, carefully selected from an initial pool of 1 billion noisy web image-text pairs. *SSTK* contains approximately 350 million cleaned data points. To ensure high quality and reduce bias and toxicity, we have applied a series of automatic filters to the sources of both datasets. These filters include, but are not limited to, the removal of NSFW content, low aesthetic images, duplicate images, and small images. In addition, to generate dense captions for ablation experiments, we used LLaVA-v1.6-Mistral-7B (Liu et al., 2024b;a; Jiang et al., 2023).

## B    MORE ABLATIONS ON PIXART-$\alpha$

**The Effect of pretrained weights as initialization**    We compare the initialization strategy by training the original PixArt-$\alpha$ with T5-XXL (Chung et al., 2022) on LensArt with different initializations: (1) train from scratch and (2) initialize from PixArt-256 checkpoints. Better initialization indeed helps. We see more structural output after 10K steps with the PixArt-$\alpha$-256 checkpoint as initialization in comparison with training from scratch. Initializing with pre-trained PixArt-256 checkpoint has 0.82 TIFA score and 0.49 ImageReward average score at the beginning. However, the scores do not improve much during training, which implies its upper limit to be close to 0.82 for TIFA.

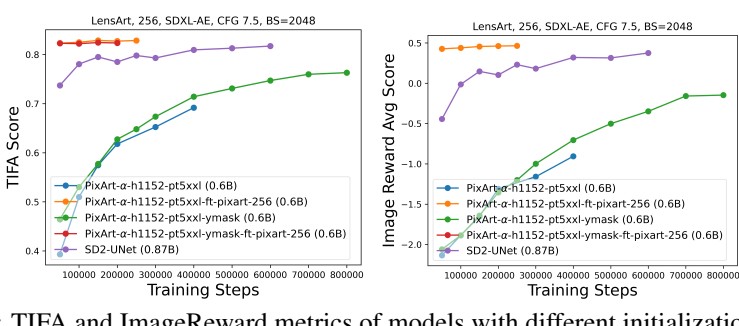

Figure 15: TIFA and ImageReward metrics of models with different initialization weights.

**The role of text encoders** We compared the impact of T5XXL and OpenCLIP-H on the convergence speed in Fig. 16. We find T5XXL (4096 token dim) takes longer time to train in comparison with OpenCLIP-H (1024 token dim), which implies longer token dimension needs more iterations to learn.

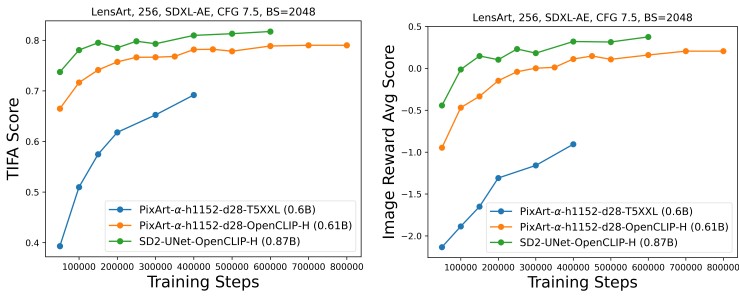

Figure 16: TIFA and ImageReward metrics of models with different text encoders.

## C QUANTITATIVE EVALUATION OF IMAGE INPAINTING AND CANNY EDGE CONDITIONING MODELS

We compare our image inpainting and canny edge conditioned models, trained using token concatenation on top of 2.3B 1K resolution U-ViT model, with BrushNet (Ju et al., 2024) and SD3-Medium-Canny-ControlNet (Esser et al., 2024a) respectively.

### C.1 BENCHMARKS

We outline the evaluation benchmarks used to assess the performance of our image inpainting and canny edge conditioning models.

**TIFA** TIFA benchmark has a set of prompts, and a set of question-answer pairs associated with each prompt. Given a prompt, the generated image is processed by a VQA model and the answers generated by it are matched against GT answers to assign a score. The scores for 4,081 prompts are averaged to obtain the final score.

- **TIFA for image inpainting:** Unlike text-to-image (T2I), the image inpainting task requires a ground truth image, a mask, and a text prompt as input. Out of the 4,081 text prompts in TIFA benchmark, 2000 are taken from the MSCOCO dataset (Lin et al., 2014), and have a corresponding image and bounding boxes associated with it. We evaluate the inpainting model on this subset of TIFA benchmark. We will refer to this modified metric as TIFA-COCO. For each prompt, we select the bounding box with largest area and convert it to a mask. Given this input, image inpainting model generates an output which is processed by the VQA to get a TIFA score. An inpainting model that can keep the unmasked details preserved while generating the masked details reliably will have a higher TIFA score, since the images generated by it will answer most questions correctly. From our findings, we observe that TIFA is a good metric for text-image alignment for image inpainting.

- **TIFA for canny edge conditioned generation**: Similar to inpainting, we evaluate canny edge conditioned model on the 2000 MSCOCO images part of the TIFA benchmark. We extract the edges from these images, and generate outputs, which are then processed by VQA to get a TIFA score. We will refer to this modified metric as TIFA-COCO.

**BrushBench** BrushBench has 600 images with corresponding inside-inpainting and outside-inpainting masks. The authors of BreshBench propose evaluating image inpainting on following metrics:

- **Image Generation Quality:** ImageReward (IR) (Xu et al., 2023), HPS v2 (HPS) (Wu et al., 2023), and Aesthetic Score (AS) (Schuhmann et al., 2022) as they align with human perception.
- **Masked Region Preservation:** Peak Signal-to-Noise Ratio (PSNR) (contributors, 2024), Learned Perceptual Image Patch Similarity (LPIPS) (Zhang et al., 2018), and Mean Squared Error (MSE) (Wikipedia contributors, 2024) in the unmasked region among the generated image and the original image.
- **Text Alignment:** CLIP Similarity (CLIP Sim) (Wu et al., 2021) to evaluate text-image consistency between the generated images and corresponding text prompts.

## C.2 TOKEN CONCATENATION INPAINTING VS BRUSHNET

We compare our 2.3B U-ViT token concatenation inpainting approach with BrushNet in Tables 2 and 3. Our approach outperforms BrushNet on all metrics except Aesthetic Score, where we achieve comparable performance.

| Model | Image generation quality | | | Masked Region Preservation | | | Text Alignment | |
|---|---|---|---|---|---|---|---|---|
| | IR $_{x\,10}$↑ | HPSv2 $_{x\,100}$↑ | AS↑ | PSNR↑ | LPIPS $_{x\,1000}$↓ | MSE $_{x\,1000}$↓ | CLIP Sim↑ | TIFA-COCO↑ |
| **Ours** | **12.82** | **27.83** | 6.47 | **32.13** | **0.72** | **18.21** | **26.50** | **89.8** |
| BrushNet | 12.64 | 27.78 | **6.51** | 31.94 | 0.80 | 18.67 | 26.39 | 88.5 |

Table 2: Performance comparison of models across various metrics on inside-inpainting masks. ↑ indicates higher is better, and ↓ indicates lower is better.

| Model | Image generation quality | | | Masked Region Preservation | | | Text Alignment |
|---|---|---|---|---|---|---|---|
| | IR $_{x\,10}$↑ | HPSv2 $_{x\,100}$↑ | AS↑ | PSNR↑ | LPIPS $_{x\,1000}$↓ | MSE $_{x\,1000}$↓ | CLIP Sim↑ |
| **Ours** | **11.01** | **29.75** | 6.55 | **29.20** | **2.13** | **4.40** | **27.30** |
| BrushNet | 10.88 | 28.09 | **6.64** | 27.82 | 2.25 | 4.63 | 27.22 |

Table 3: Performance comparison of models across various metrics on outside-inpainting masks. ↑ indicates higher is better, and ↓ indicates lower is better.

## C.3 TOKEN CONCATENATION CANNY EDGE CONDITIONING VS SD3-MEDIUM CANNY CONTROLNET

We compare our 2.3B U-ViT token concatenation canny edge conditioned generation approach with SD3-Medium-Canny-ControlNet on BrushBench evaluation set in Table 4 and on TIFA-COCO evaluation set in Table 5. Our approach outperforms SD3-Medium-Canny-ControlNet on all metrics.

| Model | Image generation quality | | | Text Alignment |
|---|---|---|---|---|
| | IR $_{x\,10}$↑ | HPSv2 $_{x\,100}$↑ | AS↑ | CLIP Sim↑ |
| **Ours** | **14.8** | **29.93** | **6.45** | **27.69** |
| SD3-Medium-ControlNet | 14.2 | 28.88 | 6.22 | 27.61 |

Table 4: Performance comparison of models across various metrics on BrushBench evaluation set. Higher values (↑) indicate better performance.

| Model | FID↓ | TIFA-COCO↑ |
|---|---|---|
| **Ours** | **5.78** | **91.9** |
| SD3-Medium-ControlNet | 5.95 | 91.1 |

Table 5: Performance comparison of models across various metrics on TIFA-COCO evaluation set. Lower FID (↓) and higher TIFA-COCO (↑) values indicate better performance.

# D MORE EVALUATION METRICS

We use two prompt sets for evaluation of text image alignment and image quality: 1) 4081 prompts from TIFA (Hu et al., 2023) benchmark. The benchmark contains questions about 4,550 distinct elements in 12 categories, including *object*, *animal/human*, *attribute*, *activity*, *spatial*, *location*, *color*, *counting*, *food*, *material*, *shape*, and *other*. 2) randomly sampled 10K prompts from MSCOCO (Lin et al., 2014) 2014 validation set, we name it MSCOCO-10K.

In addition to TIFA and ImageReward, we also provide the FID score, which measures the fidelity or similarity of the generated images to the groundtruth images. The score is calculated based on the MSCOCO-10K prompts and their corresponding images.

Fig 17, Fig 18, Fig 19 and Fig 20 are updated Figures for Fig 3, Fig 4, Fig 5 and Fig 6 with addition of FID scores.

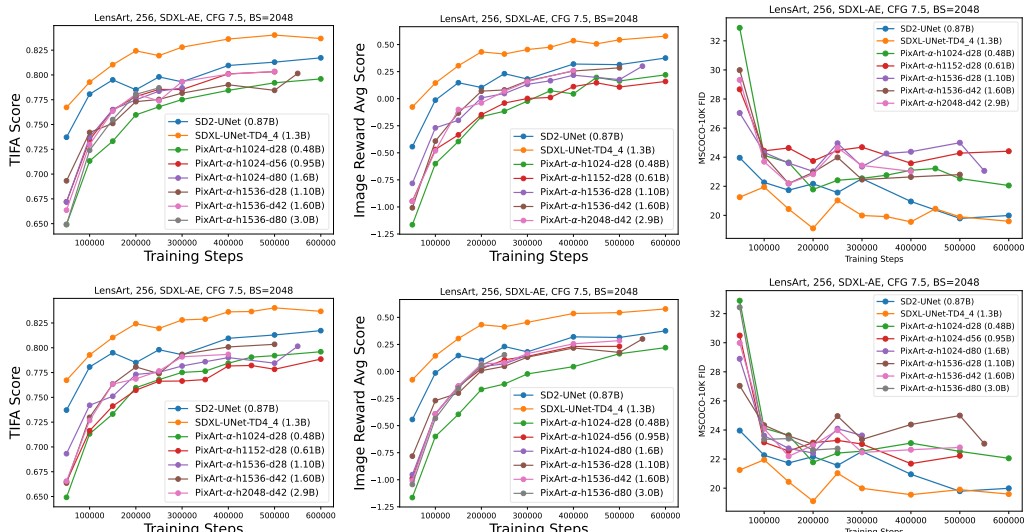

Figure 17: Scaling PixArt-$\alpha$ on the depth and width dimensions in terms of TIFA, ImageReward and FID. The first row is scaling along

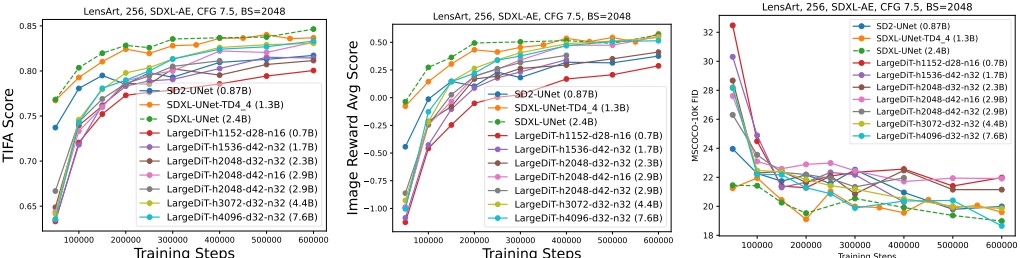

Figure 18: Scaling LargeDiT variants from 1.7B to 8B and their performance in TIFA, ImageReward and FID.

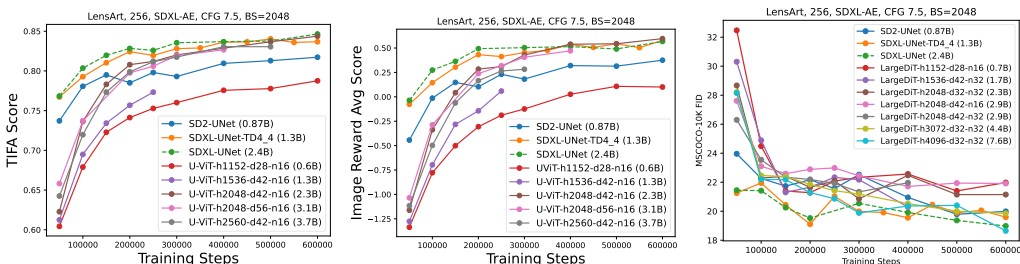

Figure 19: Scaling U-ViT along width $h$, depth $d$ and combined dimensions in terms of TIFA, ImageReward and FID.

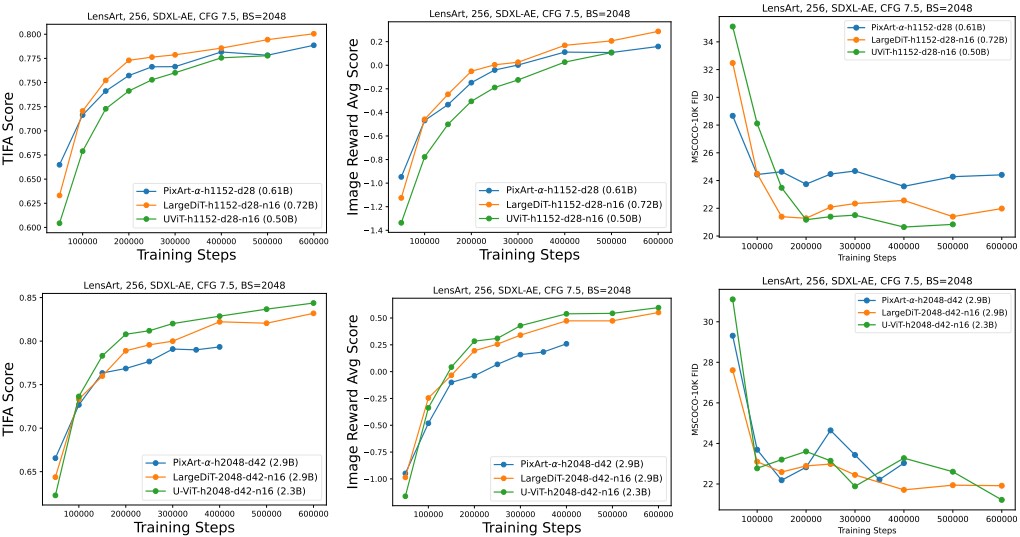

Figure 20: Comparing different DiT designs in similar architecture hyperparameters at different scales in terms of TIFA, ImageReward and FID.

