# OpenReview forum: "Efficient Scaling of Diffusion Transformers for Text-to-Image Generation"
_ICLR.cc/2025/Conference — Submitted to ICLR 2025_

### Official Review · Reviewer_F4tt · 2024-10-28

**Soundness:** 1
**Presentation:** 2
**Contribution:** 1
**Rating:** 3
**Confidence:** 2

**Summary:**

This paper study the scaling properties of various Diffusion Transformers (DiTs) for text-to-image generation, including training scaled DiTs ranging from 0.3B upto 8B parameters on datasets up to 600M images.

**Strengths:**

The research toppic is popular

The paper is well-written

**Weaknesses:**

There is not much to take in this paper.  The paper seems to be incomplet:
1) important metrics like FID and IS are not used in the paper
2) although the authors study the scalling of DITs, they didn't privide a stonger version of DIT. Hence, they didn't demenstrate the reliablity of their paper.
3) Conclusions from the paper  like:'Finetuning text encoder improves the convergence speed for UNet' are not important. Because finetuning usually improves the performance..
4) what's more, contribution from the paper:'We examined why long caption enhancement and dataset scaling help to improve the
text-image alignment performance.' seems not straitforward.  Then the authors claim that 'Thus, we conclude that captions with higher information density (i.e., not necessarily long caption length) is key to improve text-image alignment'. Similar to 3), the authors again provide useless points.

**Questions:**

What's the contribution of this paper?
Could the authors provide stronger models enhanced by conclusions from the paper?
Could the authors provide FID results?

---

> ### Author Response · Authors · 2024-11-29
> **Response to Reviewer F4tt**
>
> We sincerely thank the reviewer’s critical comments. However, we respectively disagree with the comments and would like to clarify the misunderstood part:
>
> > W1: “important metrics like FID and IS are not used in the paper”  and Q3: “Could the authors provide FID results?”
> - Inception Score (IS) is mainly used in evaluating class-conditioned image generation. Our work focuses on text conditioned image generation, so IS cannot be used here.
> - FID score is a common metric for assessing early T2I models.  However, as reported in many other works [1,2,3,4], FID is not quite reliable for accessing T2I models. We also observed it is not very stable in comparison with TIFA and ImageReward.
>  - In Appendix D of the updated draft, we calculate the FID score based on MSCOCO-10K images for each section along with TIFA and ImageReward scores. We can see the FID score has more variations in comparison with TIFA and ImageReward. We do use FID for evaluating edge-to-image model which has groundtruth images.
>
>
> > W2: “although the authors study the scalling of DITs, they didn't privide a stonger version of DIT. Hence, they didn't demonstrate the reliablity of their paper.”
>
> - We do provide stronger version of U-ViTs, i.e., a scaled U-ViT (2.3B) which surpassed the performance of original U-ViTs and other cross-attention based DiTs in much larger sizes (up to 8B). We argue that scaling is an important dimension of architecture design where the simplicity and scalability matters.
> - We show that a straightforward extension of condition tokens can provide a strong U-ViT inpainting model than outperforms SOTA inpainting model (BrushNet) and edge-to-image model (SD3-ControlNet). Specifically, we demonstrate that simple token concatenation can outperform BrushNet on image quality, masked region preservation, and text alignment for both inside-inpainting and outside-inpainting masks. More details can be found in Appendix D or https://openreview.net/forum?id=iG7qH9Kdao¬eId=7mQ5v5Rfdf.
>
> > W3: “Conclusions from the paper like:'Finetuning text encoder improves the convergence speed for UNet' are not important. Because fine-tuning usually improves the performance..”
>
> - The goal of the experiment is to verify U-ViT can implicitly act as fine-tuning text embedding during training while not actually fine-tune the text encoder, and to explain why U-ViT helps to improve the T2I alignment in comparison with cross-attention based methods. In practice, fine-tuning the text encoder during training will cost more memory and training FLOPs, so it is usually not adopted when training data is less. To the best of our knowledge, no existing works demonstrate the benefit of text encoder fine-tuning. Our experiments also demonstrate that scaled U-ViT can act as implicitly improving text representation during training.  We have updated section 4.2 with both ablations on cross-attention and self-attention models to make the narratives more clear.
>
> > W4: “what's more, contribution from the paper:'We examined why long caption enhancement and dataset scaling help to improve the text-image alignment performance.' seems not straitforward. Then the authors claim that 'Thus, we conclude that captions with higher information density (i.e., not necessarily long caption length) is key to improve text-image alignment'. Similar to 3), the authors again provide useless points.”
>
> - We investigate why long caption and data scaling provide better text-image alignment performance, specifically what helps to improve the TIFA score. We had experiments comparing different caption sources, including original caption, short synthetic caption, and long synthetic captions. As shown in the updated Fig. 12 and Fig. 14, the model trained with dataset that comes with high information density can yields higher TIFA score with the same number of training steps. To our knowledge, this is the first study about why long caption helps to improve text-image alignment score especially for TIFA, which provides guidance for future data scaling and enhancement.

---

> ### Author Response · Authors · 2024-11-29
> **Response to Reviewer F4tt (part 2)**
>
> > Q1: “What's the contribution of this paper?”:
>
>
> - Our main contribution includes:
>    * We compared major DiT variants for T2I in a controlled setting to allow fair comparison of recent DiT variants, which was not fully explored before.
>    * We demonstrated that simple U-ViT design can be effectively scaled for T2I generation. We show that a 2.3B U-ViT can surpasses the performance of similar or larger sized UNet, PixArt-alpha and LargeDiTs (upto 8B) and scales with datasets size increase and long caption enhancement.
>    * We show that the full self-attention design of U-ViT allows concatenation of mask and condition images , which shows better performance than traditional channel concatenation based approaches for inpainting and canny edge.
>    * We verified the U-ViT model can get better performance by scaling the data size enhanced with long captions. We verified captions with higher information density can yield better text-image alignment performance.
>
> - Difference with other DiTs
>   * We did not re-invent “new” architectures, but try to understand and push the limit of existing design. While recent DiT variants may have different normalization layers and PoEs, they mainly still follow the original DiT’s cross-attention design, while ignoring the most simple self-attention based model such as U-ViT, which provides much simpler block design and allows straightforward token extension.ith other DiTs
>
>
> > Q2: “Could the authors provide stronger models enhanced by conclusions from the paper?”
>
> - A SOTA T2I model is usually composed of several strong components, including VAE, denoising backbone, text encoder, training data quality and training recipe. The focus of this paper is to show the simple designed U-ViT can be scaled in following dimensions to obtain stronger performance:
>   * **backbone scaling**:  we show that properly scaled U-ViTs can yield significant performance gain over its original version (U-ViT-Huge) as well as other cross-attention based methods like SDXL, scaled PixArt-alpha and LargeDiT models up to 8B in terms of TIFA score and ImageReward. We show samples of high resolution and high quality images can be generated by the 1K resolution model in Fig 1.
>   *  **data scaling**: we show that the scaled U-ViT model can benefit from dataset size scaling (upto 600M) and long caption enhancement in terms of both TIFA and ImageReward.
>   * **token scaling**: we demonstrate that a straightforward extension of conditioning tokens can derives stronger inpainting model and edge-to-image model in comparison with SOTA BrushNet and SD3-ControlNet. We show better evaluation metrics such image quality, masked region preservation, and text alignment for both inpainting and edge-to-image model. Details can be seen in Appendix D or https://openreview.net/forum?id=iG7qH9Kdao&noteId=7mQ5v5Rfdf .
>
> We hope our replies can address the reviewer’s concern and our work could be re-evaluated.
>
> Referenece:
> - [1] Podell et al, SDXL: Improving Latent Diffusion Models for High-Resolution Image Synthesis, ICLR 2024 https://arxiv.org/pdf/2307.01952
> - [2] Jayasumana et al, Rethinking FID: Towards a Better Evaluation Metric for Image Generation, CVPR 2024
> https://arxiv.org/pdf/2401.09603
> - [3] Chong and Forsyth, Effectively Unbiased FID and Inception Score and where to find them, CVPR 2020 https://arxiv.org/pdf/1911.07023
> - [4] Parmar et al, On Aliased Resizing and Surprising Subtleties in GAN Evaluation, CVPR 2022  https://arxiv.org/pdf/2401.09603

---

### Official Review · Reviewer_9VND · 2024-11-04

**Soundness:** 3
**Presentation:** 3
**Contribution:** 3
**Rating:** 6
**Confidence:** 4

**Summary:**

This paper conducted extensive experiments on the scaling properties of various diffusion transformers for text-to-image generation. From model scaling perspective, it discovered that a pure self-attention based variant called U-ViT scales more effectively than other variants including PixArt-$\alpha$ and LargeDiT. It also observed that the self-attention nature allows U-ViT models to take extra conditions and other modalities in a straightforward manner. From data scaling perspective, it observed that U-ViT models have better scalability given a larger training data size.

**Strengths:**

- This paper conducted extensive experiments to examine the model scaling properties of different kinds of models, including UNet based ones like SD2 and SDXL, and transformer based ones like UViT, LargeDiT, and PixArt-$\alpha$.
- This paper investigated many scaling perspectives, including model size scaling, data size scaling, caption scaling, token number scaling, and so on.
- This paper delivered a message that UViT models have better scaling properties, which can provide a reference for future research.

**Weaknesses:**

- Despite the extensive study, there seem no novel technical contributions within this paper.
- The analysis regarding why long caption enhancement and dataset scaling help to improve the text-image alignment performance seems not thorough enough.

**Questions:**

- In Figure 4 left, why does U-ViT-h2560-d42-n16 perform worse than U-ViT-h2048-d42-n16?
- In section 4.2, the experiments on fine-tuning text encoders are carried to SDXL, PixArt-$\alpha$, and LargeDiT. Why not also study U-ViT models? Although there might not be significant improvement since transformer blocks in U-ViT **may** be implicitly considered as part of the text encoder which is being fine-tuned, conducting the study of U-ViT models does no harm, and can further validate the correctness of the previous reason.
- In section 6.1 about caption scaling, which model was used to compare LensArt and LensArt-Long-50%? Why not conduct a more thorough ablation with all variants of UNet based models and transformer based models, like in section 6.2?
- Since the models studied in sections 6.1 and 6.2 are not the same, the analysis in section 6.3 about “information density” seems not that convincing. For example, since in figure 13, the LensArt Long Caption consistently outperforms SSTK in terms of “the percentage of captions with matched TIFA element phrases in each TIFA element type”, can we expect that using the same model, training with LensArt Long Caption can lead to better performance improvement than SSTK?

---

> ### Author Response · Authors · 2024-11-28
> **Response to Reviewer 9VND**
>
> We appreciate the reviewer’s constructive suggestions and acknowledging the strength and contribution of our work. We hope the following clarifications can address the reviewer's concerns.
>
> > Q1: “In Figure 4 left, why does U-ViT-h2560-d42-n16 perform worse than U-ViT-h2048-d42-n16?”
>
> * We have updated the Figure. As seen in the updated Fig 5 left, both U-ViT-h2048-d56-n16 (3.1B) and U-ViT-h2560-d42-n16 (3.7B)’s performance are very close to U-ViT-h2048-d42-n16 (2.3B) in terms of TIFA and ImageReward, indicating that continue increasing the width and depth has limited improvement on the metrics with the LensArt dataset.
>
> > Q2: “In section 4.2, the experiments on fine-tuning text encoders are carried to SDXL, PixArt-, and LargeDiT. Why not also study U-ViT models? Although there might not be significant improvement since transformer blocks in U-ViT may be implicitly considered as part of the text encoder which is being fine-tuned, conducting the study of U-ViT models does no harm, and can further validate the correctness of the previous reason.”
>
> * Thanks for the suggestion, we added U-ViT experiments for further verification. We trained two versions of U-ViT models, 0.6B and 2.3B with text encoder fine-tuning enabled and disabled. We find that when U-ViT is small and shallow, fine-tuning text encoder achieves significant improvement over its non-finetuned version. While U-ViT size is large (2.3B), the text encoder fine-tuned version slows down the convergence over the frozen version.
>
> > Q3: “In section 6.1 about caption scaling, which model was used to compare LensArt and LensArt-Long-50%? Why not conduct a more thorough ablation with all variants of UNet based models and transformer based models, like in section 6.2?”
>
> Thanks for the suggestion. We originally used UNet-C128 (0.4B) for the initial experiment LensArt and LensArt-long. Previous work [Li et al, 2024](https://arxiv.org/pdf/2404.02883) has shown that the caption enhancement is orthogonal to architecture improvement. In the updated draft, we reran the experiments with the 0.6B U-ViT with LensArt original caption, long caption and SSTK caption and updated Fig. 12. We show that LensArt long yields better performance than the original caption and SSTK caption, which is consistent with information density shown in Fig.14.
>
> > Q4: “Since the models studied in sections 6.1 and 6.2 are not the same, the analysis in section 6.3 about “information density” seems not that convincing. For example, since in figure 13, the LensArt Long Caption consistently outperforms SSTK in terms of “the percentage of captions with matched TIFA element phrases in each TIFA element type”, can we expect that using the same model, training with LensArt Long Caption can lead to better performance improvement than SSTK?”
>
> Yes. We updated the Fig.12 in Section 6.1 with training curves of 0.6B U-ViT model with LensArt original caption, LensArt Long caption and SSTK caption. We can see that training with LensArt long caption leads to better performance than LensArt original caption and SSTK. The relative performance is consistent with the observed information density in Fig 14.

---

> > ### Comment · Reviewer_9VND · 2024-12-03
> >
> > Thank you for the response. However, I still have several concerns:
> >   - The updates in Figure 5 are quite confusing. It seems that the authors modified the curve for “U-ViT-h2560-d42-n16” without any explanation (comparing Figure 4 left in the original version and Figure 5 left in the updated version).
> >   - In Figure 8 (b), the comparison between "U-ViT-h2048-d42-n16 (2.3B)" and "U-ViT-h2048-d42-n16 (2.3B) | OpenCLIP-FT" presents an unexpected result: the version with the fine-tuned text encoder shows slower convergence compared to the frozen version. This counter-intuitive finding warrants further explanation.

---

> > > ### Author Response · Authors · 2024-12-04
> > >
> > > We appreciate the reviewer’s detailed examination and reply to our response, which helps us to further improve our work.
> > >
> > > > The updates in Figure 5 are quite confusing. It seems that the authors modified the curve for “U-ViT-h2560-d42-n16” without any explanation (comparing Figure 4 left in the original version and Figure 5 left in the updated version).
> > >
> > > * Sorry for missing the details on how the figure was updated. We traced the issue and identified the reason was it accidentally used smaller batch size (1024 rather than 2048) when training with only half number of nodes, which results in slower convergence in comparison with other curves. We retrained the “U-ViT-h2560-d42-n16” with the correct setting and re-plot the curve. However, the right ImageReward curve was not updated accordingly, which caused the confusion. We will fix the curve in Fig.5 right in the final version. We see both TIFA and ImageReward curves for “U-ViT-h2560-d42-n16” reach the same performance as the “U-ViT-h2048-d42-n16” version.
> > >
> > >
> > > > In Figure 8 (b), the comparison between "U-ViT-h2048-d42-n16 (2.3B)" and "U-ViT-h2048-d42-n16 (2.3B) | OpenCLIP-FT" presents an unexpected result: the version with the fine-tuned text encoder shows slower convergence compared to the frozen version. This counter-intuitive finding warrants further explanation.
> > >
> > > * Thanks for spotting this issue. We again identified that the “U-ViT-h2048-d42-n16 (2.3B) | OpenCLIP-FT” in Fig.8 (b) was using a smaller batch size (1024) and thus caused slower convergence. We reran the 2.3B U-ViT model with fine-tuned text-encoder in the batch size 2048 setting. The corrected TIFA curve slightly surpasses the frozen version in the early stage of training. The relative improvement over the frozen version is smaller in comparison with the improvement made on the 0.6B U-ViT model.  We will update the figure in the final version.
> > >
> > > We will update figures accordingly and hope the clarification can address the reviewer’s concerns.

---

### Official Review · Reviewer_V7ao · 2024-11-04

**Soundness:** 3
**Presentation:** 3
**Contribution:** 3
**Rating:** 6
**Confidence:** 3

**Summary:**

This paper conducts extensive experiments to analyze properties of different diffusion transformer variants, providing some valuable practical findings.

**Strengths:**

Extensive experiments of models with different architectures, different numbers of parameters and different dataset settings are conducted.
Given the fact that transformer-based diffusion models are important and popular in both image and video generations.
Practical findings from this paper could contribute to the community and help other researchers in future model architecture design and model training.

**Weaknesses:**

Although the paper provides some valuable experience and findings, it lacks in-depth analysis.
I appreciate the author's efforts in conducting comprehensive experiments and offering the practical findings, but only performing ablation studies based on existing model designs might be a weakness of a research paper.

In experiments, some models are not trained with the same number of steps. As a result, we can compare their performance at the early stages of training, but whether the comparison will remain the same after fully convergence is actually uncertain.

Evaluating inpainting task with TIFA and ImageReward may not be appropriate.

Only some simple qualitative results are presented for edge-to-image generation in section 5.2. More quantitive comparison with existing, related methods are suggested.

**Questions:**

Is it possible that the performance gap between fine-tuning/freezing text encoders is partially because of the incapability of the chosen text encoder? In other words, if one use a much stronger text encoder, will the performance gap become smaller?

What is the reason of many models in some figures are trained with different steps, even when they have the same architecture design?

---

> ### Author Response · Authors · 2024-11-28
> **Responses to Reviewer V7ao (part1)**
>
> We thank the reviewer for the positive and constructive comments. We are encouraged by the recognition of our contribution to the community. Below are our responses to the weaknesses and questions:
>
> Weakness:
> > W1: “only performing ablation studies based on existing model designs might be a weakness of a research paper”
>
> * The goal of the study is to understand the scalability of widely used DiT variants, i.e., we investigate whether a simple architecture can be effectively scaled along parameter size dimension and token size dimension. We show both quantitive and qualitative results that the scaled U-ViT can outperform SDXL and other DiT variants for T2I, and also enables straightforward extension for image editing.
>
> > W2: “some models are not trained with the same number of steps... but whether the comparison will remain the same after fully convergence is actually uncertain.”
>
>  * Due to the scale of the experiments, some of the curves were not complete due to computation constraints. However, we find the early convergence of the same architecture pretty indicative for predicting the final performance. Some of the small architecture variants have clear performance gap with others, which cannot be mitigated after more steps of training, so we terminated some of the jobs early. In the updated draft, we updated the figures with more completed steps.
>
> > W3: “Evaluating inpainting task with TIFA and ImageReward may not be appropriate. Only some simple qualitative results are presented for edge-to-image generation in section 5.2. More quantitive comparison with existing, related methods are suggested.”
>
> * TIFA score is a reliable metric to evaluate text-image alignment for T2I models. We find that a customized TIFA score, which we will call TIFA-COCO and is explained in more detail below, can be used to get a good image-text alignment score for image inpainting task.
>   * The TIFA benchmark has a set of prompts, and a set of question-answer pairs associated with each prompt. Given a prompt, the generated image is processed by a VQA model and the generated answers are matched against GT answers to assign a score.
>   * Unlike T2I, the inpainting task requires a ground truth image, a mask, and a text prompt as input. Out of the 4,081 text prompts in TIFA benchmark, 2000 are taken from the MSCOCO dataset, and have corresponding image and bounding boxes. We evaluate the inpainting model on this subset of TIFA benchmark, and call this metric TIFA-COCO. For each prompt, we select the bounding box with largest area and convert it to a mask. Given this input, image inpainting model generates an output which is processed by the VQA to get a TIFA-COCO score. An inpainting model that can keep the unmasked details preserved while generating the masked details reliably will have a higher TIFA-COCO score, since the images generated by it will answer most questions correctly. From our findings, we observe that TIFA-COCO is a good metric for text-image alignment for image inpainting.
>
> * Comparing inpainting model with related works on multiple metrics
>      * We compare our image inpainting task with other recent related works on more metrics than just TIFA-COCO and ImageReward. For this, we use the BrushBench evaluation set. BrushBench has 600 images with corresponding inside-inpainting and outside-inpainting masks. The authors of BreshBench propose evaluating image inpainting on following metrics:
>         * **Image Generation Quality**: Image Reward (IR), HPS v2 (HPS), and Aesthetic Score(AS) as they  align with human perception.
>         * **Masked Region Preservation**: Peak Signal-to-Noise Ratio (PSNR), Learned Perceptual Image Patch Similarity (LPIPS), and Mean Squared Error (MSE) in the unmasked region among the generated image and the original image.
>         * **Text Alignment**: CLIP Similarity (CLIP Sim) to evaluate text-image consistency between the generated images and corresponding text prompts.
>
>   * We show the results below. We compare our 2.3B U-ViT *inpainting* model trained using token concatenation against the BrushNet (https://arxiv.org/pdf/2403.06976) in the tables below. Brushnet has compared their model with a number of related works in their paper and shown superior performance. We show that we outperform BrushNet (and therefore all the related works BrushNet has compared against in their paper) on image quality, masked region preservation, and text alignment for both inside-inpainting and outside-inpainting masks.

---

> ### Author Response · Authors · 2024-11-28
> **Responses to Reviewer V7ao (part2)**
>
> Table 1. Performance comparison of models across various metrics on inside-inpainting masks. $\uparrow$ indicates higher is better, and $\downarrow$ indicates lower is better.
>
> |                          |    IR↑    |   HPSv2↑  |    AS↑   |   PSNR↑   |  LPIPS↓  |    MSE↓   | CLIP Sim↑ | TIFA-COCO↑ |
> |:------------------------:|:---------:|:---------:|:--------:|:---------:|:--------:|:---------:|:---------:|:----------:|
> | UViT-Token Concat (ours) | **12.82** | **27.83** |   6.47   | **32.13** | **0.72** | **18.21** |  **26.5** |  **89.8**  |
> |         BrushNet         |   12.64   |   27.78   | **6.51** |   31.94   |    0.8   |   18.67   |   26.39   |    88.5    |
>
>
> Table 2. Performance comparison of models across various metrics on outside-inpainting masks. $\uparrow$ indicates higher is better, and $\downarrow$ indicates lower is better.
>
> |                          |    IR↑    |   HPSv2↑  |    AS↑   |   PSNR↑   |  LPIPS↓  |    MSE↓   | CLIP Sim↑ | TIFA-COCO↑ |
> |:------------------------:|:---------:|:---------:|:--------:|:---------:|:--------:|:---------:|:---------:|:----------:|
> | UViT-Token Concat (ours) | **12.82** | **27.83** |   6.47   | **32.13** | **0.72** | **18.21** |  **26.5** |  **89.8**  |
> |         BrushNet         |   12.64   |   27.78   | **6.51** |   31.94   |    0.8   |   18.67   |   26.39   |    88.5    |
>
> * Quantitative comparison of edge-to-image generation with related works
>   * As requested by reviewer, we compare our 2.3B U-ViT edge-to-image generation model trained using token concatenation with the corresponding edge-to-image ControlNet trained on top of SD3-Medium. We compare these models on the Image generation quality and text alignement metrics from BrushBench evaluation set as well as FID and TIFA-COCO metrics on COCO images from TIFA benchmark.
>
>
> Table 3. Performance comparison of models across various metrics on BrushBench evaluation set. Higher values (↑) indicate better performance.
> |                          |    IR↑   |   HPSv2↑  |    AS↑   | CLIP Sim↑ |
> |:------------------------:|:--------:|:---------:|:--------:|:---------:|
> | UViT-Token Concat (ours) | **14.8** | **29.93** | **6.45** | **27.69** |
> |      SD3-ControlNet      |   14.2   |   28.88   |   6.22   |   27.61   |
>
>
> Table 4. Performance comparison of models across various metrics on TIFA-COCO evaluation set. Lower FID (↓) and higher TIFA-COCO (↑) values indicate better performance.
>
> |           Model          |   FID↓   | TIFA-COCO ↑ |
> |:------------------------:|:--------:|:-----------:|
> | UViT-Token Concat (ours) | **5.78** |  **0.919**  |
> |      SD3-ControlNet      |   5.95   |    0.911    |
>
>
> Questions
>
> > Q1: “Is it possible that the performance gap between fine-tuning/freezing text encoders is partially because of the incapability of the chosen text encoder? In other words, if one use a much stronger text encoder, will the performance gap become smaller?”
>  * The effect of fine-tuning text encoder is essentially improving the text representation. Yes, when a better text encoder is used, the gap between fine-tuning and freezing text encoder could be smaller as both will improve the text-image alignment performance.  In the updated draft, we add experiments comparing replacing OpenCLIP-H with stronger text encoder (T5XL and T5XXL) and compares with fine-tuned OpenCLIP-H in Fig. 8(c).

---

### Meta-Review · Area_Chair_AMoz · 2024-12-21

**Metareview:**

This work presented a new family of U-ViT for text-to-image generation. The authors exhaustively studied how the model scaling and data scaling of diffusion transformers affect the final text-to-image performance. In addition, the authors also found that self-attention based DiT outperforms cross-attention based DiT under the same settings.

One of the main strengths of this work, as pointed out by reviewers as well, is the comprehensive ablation studies on how different aspects including model size, training data scale, architecture designs for DiTs affect the final performance of text-to-image generation. The ACs also appreciate the efforts in this work.

Nevertheless, there are some drawbacks pointed out by the reviewers. First, the authors failed to present some insightful observations or analysis depsite the dense experimental results. Second, the reliabiltiy of the experiments in this work are questioned by reviewers after the rebuttal. Third, the authors studied different parts for the model while failed to present a coherent story for the submission as a whole.

After the rebuttal, this work got 3,6,6 as the final ratings, while the reviewer F4tt who gave 3 has limited experience in this domain.  To draw the final conclusion, the ACs carefully read the submission and the authors' rebutta. When checking the other two reviewers' comments, the ACs agree with both of them that this work presented extensive experiments and ablations while with limited the in-depth analysis correspondingly. One of the reviewer was also concerned about the rigriousness of the experimental settings regarding the training steps, batch sizes, etc. After the rebuttal, the reliability of the experimental result was remained as a concern.

In the light of remaining weaknesses pointed out by the reviewers, the ACs think this submission is not ready to appear in this venue, but highligh recommend the  authors reorganize their experiments, follow more reliable experimental settings, and conduct more conclusive in-depth analysis.

**Additional Comments On Reviewer Discussion:**

The reviewer discussions unfortunately did not go well - only two reviewers made the final recommendation after the discussion session. To make the final decision, the ACs read the full submission and reviewers' comments. As pointed out above, the ACs recognized the contribution of this work for conducting extensive experiments, while also shared similar concerns about the lack of novelty, and in-depth analysis as well as the reliability shown in the experiments.

---

### Decision · Program_Chairs · 2025-01-22

Reject